# Influence of Low-Intensity Ultrasound on *ε*-Polylysine Production: Intracellular ATP and Key Biosynthesis Enzymes during *Streptomyces albulus* Fermentation

**DOI:** 10.3390/foods11213525

**Published:** 2022-11-05

**Authors:** Jiahui Xiang, Mokhtar Dabbour, Xianli Gao, Benjamin Kumah Mintah, Yao Yang, Wenbin Ren, Ronghai He, Chunhua Dai, Haile Ma

**Affiliations:** 1School of Food and Biological Engineering, Jiangsu University, 301 Xuefu Road, Zhenjiang 212013, China; 2Institute of Food Physical Processing, Jiangsu University, 301 Xuefu Road, Zhenjiang 212013, China; 3Key Laboratory of Food Processing and Quality Control, College of Food Science and Technology, Nanjing Agricultural University, Nanjing 210059, China; 4Department of Agricultural and Biosystems Engineering, Faculty of Agriculture, Benha University, Qaluobia P.O. Box 13736, Egypt; 5Council for Scientific and Industrial Research—Food Research Institute, Accra P.O. Box M20, Ghana

**Keywords:** *Streptomyces albulus*, *ε*-polylysine, ultrasound-assisted fermentation, intracellular ATP, *Biosynthesis* key enzymes

## Abstract

The effect of low-intensity sonication treatment on cell growth, *ε*-polylysine (*ε*-PL) yield and its biological mechanism were investigated, using a 3-L-jar fermenter coupled with an in situ ultrasonic slot with a *Streptomyces albulus* strain SAR 14-116. Under ultrasonic conditions (28 kHz, 0.37 W cm^−2^, 60 min), a high biomass of SAR 14-116 and concentration of *ε*-PL were realized (i.e., they increased by 14.92% and 28.45%, respectively) when compared with a control. Besides this, ultrasonication increased the mycelia viability and intracellular ATP as well as activities of key enzymes involved in the *ε*-PL biosynthesis pathway, resulting in an improvement in the production of *ε*-PL. Data on qRT-PCR revealed that ultrasonication also affected the gene expression of key enzymes in the *ε*-PL biosynthesis pathway, including *ε*-PL synthetase (PLS). These outcomes provided the basis for understanding the effects of ultrasound-assisted fermentation on the stimulation of metabolite production and fermentation procedure in a fermenter.

## 1. Introduction

As an effective non-thermal physical (i.e., green) processing technique, ultrasonication is extensively applied in the food industry [1,2]. Ultrasound has also been applied to food microbial fermentation to enhance productivity and process efficiency in the food industry [3]. In general, ultrasound has dual effects on microorganisms. High frequencies (2–10 MHz), which can denature enzymes or break cells, has been typically used as a nondestructive analytical technique for monitoring fermentation processes [4]. At present, acoustic cavitation is considered a concept of deactivation by high-intensity ultrasound. The form-grow-collapse of cavitation/sonication bubbles in aqueous media causes powered effects (e.g., microstreaming, shear strain and shock waves) and sonochemical reactions (including unstable atoms and hydrogen peroxide, H_2_O_2_), resulting in the damaging or disordering of bacterial cells [5,6,7]. On the other hand, low frequency (20–100 kHz) treatment has a positive impact, enhancing the mass transfer rate of gas and liquid nutrients in microbial fermentation processes without damaging the cells [8,9]. An aptly applied ultrasound treatment has the potential to enhance the bioprocesses and/or productivity of several bacteria, yeasts, actinomycetes and filamentous fungi [8]. The low-intensity ultrasound mechanically stimulates a physiological response in microorganisms. The treatment of ultrasound causes pressure changes, which increase the cellular stress and induce an excessive metabolism of cell proliferation to produce more cells for resisting adverse living conditions [10,11,12]. Additionally, ultrasonication is noted to improve fermentative processes [8] and thus, has beneficial application in fermentation science [13].

As a food-grade preservative, cationic biopolymer *ε*-polylysine (*ε*-PL) is an antimicrobial agent, which is very effective against spoilage organisms and food pathogens [14]. In the past few decades, *ε*-polylysine and its derivative, *ε*-polylysine hydrochloride, have been used in various food industrial applications in Asia, because they are biodegradable, soluble in water, edible and non-toxic [15], especially for *Salmonella*, *Escherichia coli* and other Gram-negative bacteria, which are not easily controlled by other natural antibacterial agents. *S. albulus* is noted to be an excellent source for *ε*-polylysine [16]. Several studies have been conducted to evaluate *ε*-polylysine production through fermentation-process regulation or the optimization of nutritional conditions [17,18,19,20,21], including fermentative production by the optimization of the culture medium, immobilized cell fermentation, double carbon-source fermentation and pH-regulated fermentation.

However, the low yield of *ε*-PL produced in such fermentation processes remains a major challenge to the extensive use of this natural antimicrobial agent and highly functional material. The beneficial effect of low-intensity ultrasound has been demonstrated in multiple microorganisms, including *Bacillus subtilis*, *Candida tropicalis* and *Saccharomyces cerevisiae* (referencing the stimulation of microbial cell proliferation, see [1,5,10,11,13,14,22]). Additionally, based on the influence on membrane permeability and enzymatic activities, we (in our previous study) developed an efficient ultrasound-stimulation strategy for promoting microbial fermentation and metabolite production. The data suggested that the biomass of *S. cerevisiae* increased by 127.03% under optimal conditions of low-intensity ultrasound [11]. The cell-membrane permeability was affected by the change in extracellular protein, nucleic acid (biopolymer) and fructose 1, 6-diphosphate (FDP) contents. Likewise, Zhang et al. [13] founded that ultrasound at a fixed frequency of 23 kHz significantly promoted the metabolism yield of ethanol and the content of *β*-phenylethanol and other metabolites, such as esters. To further explain the mechanism of ultrasound treatment for the enhancement of ethanol output, a 7.5 L fermentation tank connected with six-frequency (6f) ultrasonic equipment was used by He et al. [22]. Results indicated that ultrasound increased the activities of three main enzymes (hexokinase, phosphofructokinase, pyruvate kinase), which catalyzed three irreversible reactions in the glycolysis metabolism of ethanol biosynthesis, and an accelerated glucose consumption contributed to increasing the rate of ethanol production. Using sweeping frequency pulsed ultrasound (SFPU) in the pretreatment of RP (rapeseed protein) prior to proteolysis, it was found that ultrasound improved the enzymolysis efficiency by enhancing the hydrolysis rate due to the unfolding of molecular conformation and the secondary structure of proteins [23]. The physical mechanism of ultrasound-induced and enhanced enzymatic hydrolysis is not only related to the changes in molecular conformation and the microstructure of proteins, but also the increase in enzyme–substrate affinity and reaction velocity induced by intense micro-convection generated during sonication [24]. 

Nevertheless, most of the previous studies on the mechanism of ultrasonic-assisted fermentation, to date, have been carried out with the objective of promoting cell proliferation and increasing cell-membrane permeability. Even so, the studies on the influence of ultrasound on metabolic processes are still limited and insufficient. The present study, therefore, aimed at developing an efficient ultrasonic strategy to promote the synthesis and metabolism of *ε*-polylysine in a 3-L-jar fermenter coupled with an in situ ultrasonic slot. The biochemical indices and kinetic parameters during the fermentation of *S. albulus* have been investigated to verify the *ε*-PL production characteristics. Moreover, intracellular ATP levels and the activity and transcription level of key enzymes such as hexokinase (HK), glucose-6-phosphate dehydrogenase (G6PDH), pyruvate kinase (PK), aspartate kinase (ASK), aspartate aminotransferase (AAT) and *ε*-polylysine synthase (PLS), which were involved in the *ε*-PL synthesis pathway, were also studied to establish the mechanism promoting the fermentation of *S. albulus* by low-intensity ultrasound.

## 2. Materials and Methods

### 2.1. Microorganisms and Culture Media

*Streptomyces albulus* SAR 14-116 screened through mutagenesis (ARTP) [25], which is a mutant from an original *S. albulus*—CICC 11022 strain (purchased from China Center of Industrial Culture Collection (CICC)), was used in this study. The agar slant (BTN) medium was used as solid culture and the pre-culture medium (M3G) was used as seed medium [26,27]. The culture media, as regards the fermenter cultures, were prepared as outlined by Zeng et al. [19]. A glucose–glycerol mixed carbon source was fundamentally used to enhance *ε*-poly-L-lysine productivity (since it accelerates cell growth). The fermentation medium was composed of mixed carbon source (30 g/L glucose + 30 g/L glycerol) and beef extract 10 g, (NH_4_)_2_SO_4_ 10 g, MgSO_4_·7H_2_O 0.8 g, KH_2_PO_4_ 4 g and FeSO_4_·7H_2_O 0.05 g per liter, and the pH was adjusted to 6.8 using NH_4_OH (12.5%) before sterilization (121 °C, 20 min).

### 2.2. Ultrasonic Treatment of S. albulus and Fermentation of ε-Polylysine

A 3-L-jar fermenter (BioFlo/CelliGen 115, Eppendorf China Limited, Shanghai) was used in the fermentation process with 1.5 L working volume (Figure 1). Multi-frequency scanning slot ultrasound (WKS300/6S) was designed by our research group (Jiangsu university, China) and manufactured by Jiangda Wukesong Biotechnology Co. (Zhenjiang, China). This equipment has/generates several frequencies (20, 23, 25, 28, 33, 40 kHz) at a maximum power output of about 300 W, and the rated power of each generator is 50 W [13]. For use as sonobioreactor, the sonic chamber (160 mL) of the hex-frequency slit ultrasound was connected with the 3-L-jar fermenter through sterile silicone tubing as shown in Figure 1. The temperature of the samples was controlled during the treatment by a liquid circulating system and a water bath connected to the equipment. Moreover, an external peristaltic pump was employed to control the flow rate of the fermentation broth (50 mL/min) during the ultrasound treatment. The broth taken from the bottom of the fermentation was recirculated continuously with ultrasonic treatment through the ultrasonic chamber for one hour using the pump in a sterile environment, and then the broth flowed back into the upper part of the fermenter. All the steps of ultrasound-assisted fermentation were carried out with the recirculation of the broth through the sonic chamber.

Pre-cultured seed (120 mL, 24-h seed culture) was inoculated into 1.5 L sterilized fermentation medium with an initial pH of 6.8. Under the conditions of 0.05 MPa pressure, 5.0 SLPM ventilation, temperature (30 °C) and agitation (200 r/min), the level of dissolved oxygen (DO) dropped as expected (from 100% to 10% or less), monitored with a DO electrode (Mettler Toledo ISM). A Mettler Toledo ISM pH electrode was used to determine changes in pH during cultivation. Afterward, pH was maintained at 4.0 by automatically adding NH_3_ solution (12.5%, *v*/*v*) to the culture broth until end of the cultivation. 

After 30 h of inoculation, the fermented broth was ultrasonically treated under the conditions of 28 kHz, 280 W/L and 0.37 W cm^−2^ for 1 h. These conditions were chosen based on preliminary experiments, including the screening of ultrasound frequency, power density and treatment time. Samples were collected after each treatment for evaluation/analyses. Suspension (cells) flowing through the reaction chamber of ultrasonic equipment with no ultrasonic treatment was used as control. 

### 2.3. Establishment and Assessment of Fermentation Kinetic Models 

The original and mutant strains were subjected to activated culture and then transferred to 3-L-jar fermenter. The fermentation culture and ultrasonic treatment were carried out under the fermentation parameter conditions (detailed in Section 2.2). For the control group, the sample was treated in same manner (as described) but without sonication. The samples of mycelium and fermentation broth were picked for measuring biomass and *ε*-polylysine production, respectively, every 12 h within 120 h. The fitted equations were predicted and compared to illustrate the variations in the fermentation kinetics of biomass and *ε*-PL production with time. The logistic model is a typical S-shaped curve, indicating the exponential growth of bacteria. The logistic equation was used as an alternative empirical function in this study. The growth pattern of *S. albulus* meets the basics of logistic regression [28]. A custom function (Equation (1)) was used to conduct non-linear fitting (least squares) for the biomass. Previous studies have shown that the formation of *ε*-PL is partially coupled with cell growth according to the relationship between the formation of *ε*-PL and cell growth [28]. Our initial data indicated the synthesis of *ε*-PL showed a certain synchronization with cell/microbial growth at early stages of fermentation, but after the microbial/cell growth reached a maximum, the *ε*-PL could still be synthesized. Therefore, a kinetic model (Luedeking–Piret) of product formation was developed to describe the *ε*-PL production (Equation (2)).
(1)$$\;X=\frac{X_{0}X_{m}\exp\left(\mu_{m}t\right)}{X_{\max}-X_{0}{+X}_{0}{\exp(\mu }_{m}t)}$$
where *X* is dry cell weight (mg/mL), *X_0_* is the initial dry cell weight (mg/mL) and *μ_m_* is maximum growth rate (mg/h).
(2)$${P=P}_{0}{-\alpha \;X}_{0}+\alpha \frac{X_{0}X_{\max}\exp\left(\mu_{m}t\right)}{X_{\max}-X_{0}{+X}_{0}\exp\left(\mu_{m}t\right)}+\beta \frac{X_{\max}}{\mu_{m}}ln\frac{X_{\max}-X_{0}{+X}_{0}\exp\left(\mu_{m}t\right)}{X_{\max}}$$
where *P* is *ε*-PL concentration (mg/mL), *P_0_* is the initial *ε*-PL concentration (mg/mL), *t* is fermentation time (h), *α* (product formation constant) linked to cell growth and *β* is a production constant related to the biomass of *S. albulus*. The parameters of the kinetic model of cell growth and product generation were calculated from the experimental data using a nonlinear curve fitted with SPSS software (version 17.0 for Windows, SPSS, Inc., Chicago, IL, USA).

### 2.4. Assay of Mycelia-Viability Staining 

The morphology of the mycelium and the distribution of living cells in the bacteria can reflect the survival status of the cells during *ε*-PL fermentation process. To investigate the effect of sonication on cell viability under the optimal ultrasonic conditions during the fermentation of *S. albulus*, the mycelial viability (of the microbe) was detected by a green fluorescent probe [5-(6)-carboxyfluorescein diacetate succinimidyl ester (CFSE)] combined with red fluorescent probe [propidium iodide]. CFSE is a membrane-penetrating green fluorescently labeled dye [29]. Studies have confirmed that CFSE-labeled cells are characterized by stable binding. Propidium iodide (C_27_H_34_I_2_N_4_, PI) is a fluorescent dye, which penetrates damaged/dead cells(and not live cells) to label DNA. Therefore, when the dyes were used, bacteria with intact/undamaged cell membranes appear fluorescent green, whereas those with damaged/stressed membranes appear red.

Culture samples of the ultrasound and the control groups were obtained at a different time, as described in the Section 2.2. The cells were centrifuged (10 min, 4500 r/min) and washed with sterile water. The mycelium pellet was re-suspended using 100 μL phosphate buffered saline (PBS). The strains (2 in all) were prepared and mixed (1:1, *v*/*v*). An equal amount (20 μL) of the strain mixture and culture samples were mixed (on a clean slide) and kept in the dark (15 min). Images were captured with a fluorescence microscope (202-XD; COI Co., Ltd., Shanghai, China).

### 2.5. Assay of Respiration Activity

The respiration rate of sonicated and untreated *S. albulus* was determined as outlined by Bai [30] with some modifications. The changes in the dissolved oxygen of the respiration rate in the ultrasonic treatment group compared with the initial respiration rate were measured when the system was stable. The respiration rate of different groups was calculated as follows:(3)$$R=\frac{\;O\times V\;}{\;T\;\times M}$$
where *R* is respiration rate (mmol/g·h), *O* is the reading of dissolved oxygen meter (mg/L), *V* is the volume of reaction liquid (mL), *T* is reaction time (min) and *M* is mycelium biomass of the cells in the reaction liquid (g).

The promoting ratio of respiration activity in different groups was calculated as follows:(4)$$I_{R}\;(\%)=\frac{R_{i}-R_{0}}{R_{0}}\times 100$$
where *I_R_* is respiratory promotion rate (%), *R_0_* is the initial respiration rate (mmol/g·h) and *Ri* is the respiration rate of the ultrasound-treatment group (mmol/g·h).

### 2.6. Assay of Key Enzyme Activities in ε-PL Biosynthesis Pathway

The control samples and the sonicated ones under the same ultrasound treatment conditions (Section 2.2) were collected during the process of *ε*-PL biosynthesis at a 12 h interval. All procedures for cell-extract preparations were carried out at 4 °C. Mycelia were obtained by centrifugation (4000 r/min 10 min) and then washed twice with PBS for enzyme and cofactor assay. The crude enzymes from *S. albulus* were extracted/isolated and the activity of hexokinase (HK), pyruvate kinase (PK), glucose-6-phosphate dehydrogenase (G6PDH) and aspartokinase (ASK) was determined by suitable assay kits (Product Nos. A077-3-1, A076-1-1, S0189, MAK095; Nanjing JB Institute, Nanjing, China; Beyotime Institute of Biotech, Shanghai, China; Sigma-Aldrich, Inc., St. Louis, MO, USA) based on manufacturer’s guidelines/instructions. The protein (macromolecule) content of the extracts was evaluated with a Super-Bradford Protein Assay (SBPA) Kit (P0060FT; Beyotime Inst. of Biotech, Shanghai, China) following the manufacturer’s specification.

### 2.7. Assay of Intracellular ATP in ε-PL Biosynthesis Pathway

For the ATP assay, the samples were analyzed by HPLC (Waters 1525, Milford, MA, USA) using a Waters SunFire C18 (250 × 4.6 mm, 5.0 μm) column at 25 °C and a spectrophotometer at 254 nm. The mobile phase contained: 10% methanol, 90% 0.02 M K_2_HPO4 and KH_2_PO4 buffer (*v*/*v* = 1:1), then its pH was adjusted to 6.0 using H_3_PO_4_. Flow rate was kept at 0.8 mL min^−1^ and the volume (injected) was set at 10 μL. 

### 2.8. qRT-PCR Assay for the Identification of the Transcription Levels of Key Enzymes in ε-PL Biosynthesis Pathway

RNA of cultured samples was extracted using a MiniBEST Universal RNA Extraction Kit. The yield and integrity of RNA were examined using a spectrophotometer and gel electrophoresis. Reverse transcription assays were conducted with TransScript First-Strand cDNA Synthesis Super MIX for qPCR. The reverse transcription system was as follows: 0.5 μg RNA, 2 μL of 5×TransScript All-in-one SuperMix for qPCR and 0.5 μL of gDNA Remover, in a total volume of 10 μL. The reaction program was devised for 15 min (42 °C), and 5 s at 85 °C. The 10 μL RT reaction was subsequently diluted (× 10) in nuclease-free H_2_O and held at −20 °C. The transcription profiles of gene *pk*, *cs*, *pls*, *aat*, SAZ_18790, SAZ_24700 and SAZ_28490 were undertaken by qRT-PCR with qRT-PCR kit mixing 2×PerfectStartTM Green qPCR SuperMix (5 μL), cDNA (1 μL), forward primer (0.2 μL), reverse primer (0.2 μL) and nuclease-free H_2_O (3.6 μL). Reactions were incubated (94 °C, 30 s) in a 384-well optical plate (Roche, Swiss), followed by 45 cycles (94 °C, 5 s, 60 °C, 30 s). The expression levels of mRNAs were normalized and computed using the 2^−ΔΔCt^ method for relative quantification with *hrdB* as the reference gene. 

### 2.9. Biomass, NH_4_^+^-N and ε-PL Concentration Analyses

Samples were taken (12-h intervals) from the fermenter for analysis of the biomass (DCW−dried cell weight,) and the concentration of ammonia nitrogen (NH_4_^+^-N) was assessed as described by Chen et al. [18]. The portion of colorimetric change from the oxidation of residual reduced sugar (RRS) was determined by a DNS test [31]. The concentration of *ε*-PL was examined according to the protocol of Cheng et al. [32]. 

### 2.10. Statistical Analysis

All processing treatments were carried out in triplicate and the results were expressed as mean ± standard deviation (SD). An analysis of variance (ANOVA) was performed to compare the effects of different treatments (processed and unprocessed samples) using SPSS (version 17.0 for Windows, SPSS, Inc., Chicago, IL, USA) software. A *p*-value of less than 0.05 was defined as significant difference.

## 3. Results and Discussion

### 3.1. Effect of Ultrasonication on the ε-Polylysine Yield during Fermentation

Following ultrasonic treatment (28 kHz, 0.37 W cm^−2^ and 60 min) at a cultivation time of 30 h, the cell growth and *ε*-PL production profiles were measured. Figure 2 shows the profiles of sugar consumption, cell growth, pH, DO, and *ε*-PL accumulation of SAR 14-116 following sonication. The cell growth rate was accelerated over 12–36 h, and the dissolved oxygen dropped sharply because of the larger oxygen demand for cell growth. By consuming glucose and NH_4_^+^-N, the maximal value of cell mass was obtained at 36 h culture time. The biomass of *S. albulus* increased by 14.42% compared with the untreated sample. In reference to the control, the promotion rate of *ε*-polylysine yield by *S. albulus* after ultrasound treatment at this stage reached the maximum (69.45%). Although the biomass of *S. albulus* does not significantly increase after entering the *ε*-PL synthesis stage, it still needed to consume a large amount of glucose and dissolved oxygen from 30 h to 36 h of cultivation. These findings suggest that the glycolytic pathway, electron-transport chain and TCA cycle were still active [33]. The volume of *ε*-PL produced continued to increase (at a constant rate) following sonication for 60 min, which may be associated with the limited rupture of mycelia clusters induced by ultrasonication, improving the cell utilization of nutrients and cell biomass [34]. After that, the production rate (of *ε*-polylysine) gradually decreased. The reason for this could be linked to the fact that the substrate was consumed subsequent to the culture time of 48 h. Although the growth rate of cell viability and cell biomass decreased, the synthesis of *ε*-PL was not inhibited. Furthermore, the pH value declining from 5.4 to 3.9 shortened by 12 h, which showed ultrasound treatment promoted the rate of acid production. This phenomenon may be linked to cavitation induced by ultrasonication to promote the transfer of gas-liquid mass and glucose consumption to accelerate the secretion of organic acids, including lactic acid, citric acid, acetic acid, etc. Chen et al. [18] have demonstrated that the production of *ε*-PL is affected by pH value. The low pH environment is more favorable for the synthesis of *ε*-PL. At 72 h fermentation time, the content of *ε*-polylysine in the ultrasound group increased by 27.31% in comparison with the non-sonicated group. After 124 h fermentation, the biomass of SAR 14-116 and yield of *ε*-polylysine after ultrasonic treatment were 4.66 g/L and 2.69 g/L, which increased by 14.92% and 28.45%, respectively (Figure 2). Such observations suggest that the ultrasonic treatment aided the process of fermentation to produce *ε*-polylysine.

### 3.2. Fermentation Kinetics of ε-PL

The synthesis of *ε*-PL, as described in a previous study [25], had some similarity/agreement with cell growth at the early stages of fermentation. To analyze the kinetics with respect to the influence of sonication on *ε*-PL production and cell growth, the synthesis parameters of cell growth (*α*) and biomass (*β*) were quantified, depending on the results in Figure 3. Equations (5) and (6) represent the biomass of cells for SAR 14-116 with (A) and without (B) sonication treatment, respectively. Equations (7) and (8) denote the *ε*-PL yield for SAR 14-116 with (C) and without (D) treatment, respectively. Fitting curves of the kinetics of cell growth and product generation are displayed in Figure 3.
(5)$$X=\frac{11055{.9e}^{0.1118t}}{7291{.2+1289e}^{0.1118t}}$$

With: *X*_0_ = 1.2886, *X*_max_ = 8.5798, *µ*_m_ = 0.1118, *R*^2^ = 0.969.
(6)$$X=\frac{10246.3e^{0.1217t}}{7964.7+1127e^{0.1217t}}$$

With: *X*_0_ = 1.127, *X*_max_ = 9.0917, *µ*_m_ = 0.1217, *R*^2^ = 0.980.
(7)$$P=\frac{1477.1e^{0.1118t}}{7291.2+1288.6e^{0.1118t}}+64.4ln(0.8498+0.1502e^{0.1118t})+0.0066$$

With: *μ_m_* = 0.1118, *α* = 0.1336, *β* = 0.839177, *R*^2^ = 0.996.
(8)$$P=\frac{1342.3e^{0.1217t}}{7964.7{+1127e}^{0.1217t}}+180.8ln(0.8760+{0.1240e}^{0.1217t})\;+0.0279$$

With: *μ_m_* = 0.1217, *α* = 0.131, *β* = 2.419798, *R*^2^ = 0.997.

The model prediction (as exhibited in Figure 3) was in good agreement with the results, suggesting that the fitted model well represented the regularity of *ε*-PL synthesis in liquid fermentation of SAR 14-116 with (C) or without (D) sonication treatment over time. After the values of *β* and *μ_m_* for two groups were determined, the outcomes proved that ultrasonication improved mycelia growth compared to the untreated/control group. The product constant associated with SAR 14-116 cell growth of the ultrasonic group model increased by 2.88 times when compared with the control. The values of *β* variations suggested that sonication treatment accelerates the material metabolism in the fermentation process. The values of the model parameters (test and control experiments) (Figure 3), were appropriate for predicting the experimental process of production of secondary metabolites of *ε*-PL. The values of *β* variations suggested that sonication treatment accelerates the material metabolism in the fermentation process. The enhanced kinetics of metabolic reactions may possibly be explained by the increase in the substrate affinity of enzymes as well as a greater resistance to substrate inhibition [35]. As a consequence, the metabolic rate of SAR 14-116 increased following sonication, which may be one of the reasons for the increase in the production of the *ε*-PL.

### 3.3. Effect of Ultrasonication on Mycelia Viability and Metabolic Activity of S. albulus

To analyze the impact of the ultrasonic time (30 and 60 min) on samples, mycelia viability was observed under a fluorescence-inverted microscope using viability staining with PI and CFSE (Figure 4). Results showed that the dead cells appeared in the central core of the mycelium and occupied a large proportion/percentage of the control group (Figure 4A,C). One can infer from this that the major fraction of the hyphae was dead, which may be due to the inhibited nutrient and oxygen supply to the internal hyphae. However, the number of dead hyphae in the cells stimulated by ultrasound treatment reduced compared with control (Figure 4B,D) and the external border of active hyphae grew at a reasonably fast rate. These results suggested that cells exposed to low frequency sonication became highly viable with no negative influences on their physiology [36]. Moreover, the respiration activity of mycelia in sonicated samples (for 30 min ultrasonic treatment) increased by 12.74% over the control, as shown in Table 1. The mechanical effect induced by low intensity ultrasound is known to improve mass transfer, and the generated intense micro-turbulence (during sonication) enhances oxygen uptake in fermentation mixture from the atmosphere above the liquid surface [37], which is in agreement with our results. The cells might accelerate the metabolic rate of nutrients, suggesting that more ATP would be supplied for the function of *ε*-PL synthesis by respiration. Additionally, the respiration rate of SAR 14-116 with 60 min ultrasound treatment did not change significantly, suggesting that ultrasound may not have inhibited the respiratory metabolism. These findings demonstrated that ultrasound treatment affects mycelia viability and further controls the metabolic activity of *S. albulus* SAR 14-116.

### 3.4. Effect of Ultrasonication on Intracellular ATP Levels of S. albulus

Compared with an unsonicated sample, the concentration of intracellular ATP was significantly increased after ultrasonic treatment (Figure 5; *p* < 0.05). The optimal sonication time was 60 min. In this condition, sonication improved the intracellular ATP rapidly with a significant increase of 81.21%, and the maximum value (10.48 μmol/g) was observed at 30 h of cultivation. The concentration of intracellular ATP increased to 90.63% in comparison with the control at 36 h, which was up to 14.73 μmol/g. This may be explained by the fact that there may have been little intracellular ATP available/used for cell growth; therefore, most intracellular ATP was accumulated as a co-factor for Pls in *ε*-PL biosynthesis [33]. Such observations indicate that the catalytic function of Pls is controlled by intracellular ATP. The high levels of ATP are essential for full enzymatic activity during the synthesis of *ε*-PL.

### 3.5. Effect of Ultrasonication on Key Enzyme Activities of S. albulus

To elucidate the stimulation of *ε*-PL production and process intensification following sonication, the activity of key enzymes of SAR 14-116 was quantified at 24 h (prophase of *ε*-PL synthesis stage) and 36 h (mid-late of *ε*-PL synthesis stage). The effect of ultrasonic treatment (28 kHz, 0.37 W cm^−2^ and 280 W/L) on the content of key enzymes related to *ε*-PL synthesis in fermentation broth is exhibited in Figure 6. 

In Figure 6, it is shown that the activity of the HK enzyme did not increase immediately after ultrasonic treatment, but it increased 4.61-fold relative to a nonsonicated sample at 36 h, reaching 19.91 × 10^−4^ U/mg protein. This significant increase indicates that the ultrasonic treatment may have increased the intensity of the glycolytic pathway, resulting in an increase in glucose consumption (Figure 2). As demonstrated by Sinisterra [34], intense microturbulence produced by ultrasonic treatment causes conformational alterations in the secondary enzyme structure that improve the kinetics and activity of intracellular enzymes involved in metabolism. Notably, ultrasonication enhanced the activity of PK over the control as displayed in Figure 6C. The enhanced PK enzyme activity may have provided more carbon skeletons for the TCA cycle from 30 h to 33 h, indicating that more carbon skeletons are supplemented to synthetic precursors for other metabolites, remarkably for the yield of oxaloacetate, the precursor of aspartate consumed in L-Lys biosynthesis. The PK level in the ultrasound group decreased at 36 h of cultivation, but was still 2.49- and 2.21-fold higher than the control at 30 h and 36 h, respectively. However, the PK enzyme activity of the control group declined from 30 h of cultivation. 

In contrast, the G6PDH enzyme activity of the ultrasound-treatment group decreased over the untreated sample (Figure 6B). At 33 h, the G6PDH enzyme activity of the ultrasonic treatment group decreased to 0.99 × 10^−4^ U/mg protein, which was 1.17 times lower than the control. The G6PDH enzyme activity decreased by 42.55% compared with the control group at 36 h. This showed that ultrasound may reduce the enzyme activity of G6PDH. The reduction in the activity of G6PDH in the ultrasonic treatment group implied that the reducing synthetic capability of equivalent NADPH may have altered the carbon flux of the PPP pathway, which hindered the synthesis of erythrose used for cell growth, which agrees with the observations of cell growth (Section 3.2). More carbon skeletons were used for the L-lysine biosynthesis pathway rather than for cell growth (PPP). The phenomenon of metabolic shift is probably caused by a reduction in pH as outlined by Zeng [19]. The decrease in the G6PDH activity of the ultrasound-treated sample was in strong agreement with the findings of other researchers [38]. The inactivation was not due to thermal and/or cavitation inactivation but rather acoustic microstreaming. In addition, the reduction in PK activity at 36 h may have stimulated the phenomenon of substrate inhibition due to accumulation of intracellular ATP (Figure 5), which triggered an inhibitory effect on PK, resulting in a decrease in the supply of ATP used for cell growth [39]. Most of the ATP produced was not utilized for cell growth, but instead accumulated and used as a co-factor of Pls in *ε*-PL biosynthesis [33]. Indeed, the level of accumulated ATP was obtained when *ε*-PL was generated (Figure 5).

AAT significantly increased when cell growth reached a stationary phase. The activity of the AAT enzyme increased by 81.21% and reached 28.80 × 10^−4^ U/mg at 36 h (Figure 6D). The increase in AAT-enzyme activity increased the metabolic flux from the oxaloacetate anaplerotic metabolic pathway into the DAP pathway [40], which promoted the synthesis of the precursor substance L-Lys of *ε*-PL. As a result, the intracellular L-Lys concentration increased. Presumably, the improvement of key enzyme activity might be a reason for the increased *ε*-PL yield.

### 3.6. Effect of Ultrasonication on the Expression of Key Enzymes and Genes of S. albulus

Based on the qRT-PCR analysis, the transcription of genes related to *ε*-PL anabolism (for ultrasound treated and control) was analyzed and compared. The outcome confirmed that the transcription levels of *ask* and *hrdD* decreased and *pk*, *aat*, *pls*, SAZ_24700 and SAZ_28490 increased significantly (*p* < 0.05) following sonication (with reference to the untreated group) (Figure 7). These results were consistent with the observations of the assays of key enzyme activity. Our results have shown that sonication could enhance the enzyme activity of PK, CS and AAT from 1.34- to 1.69-fold when compared to the control. Inferring from the analysis of qRT-PCR, this increase was caused by the upregulated expression of *pk*, *cs* and *aat*. It is speculated that the expression of *pk*, *cs* and *aat* increased the transcription of HK, PK and AAT, which is considered to possibly cause the enhancement of the metabolic intensity of the glycolysis pathway, TCA cycle and DAP pathway. It is noteworthy that the expression level of *pls* encoding was upregulated 4.58 times by *ε*-PL synthetase (Pls) (Figure 7). The improvement in *pls* expression may have positively influenced the *ε*-PL synthesis, which agreed with the observation of the increased concentration of extracellular *ε*-PL caused by sonication. The increased content of intracellular ATP and the relative expression of *pls* may have jointly promoted the synthesis of *ε*-PL and consequently increased its production.

However, the expression of *ask* and *hrdD*, which encode ASK and bind with the promoter of the *pls* gene decreased (Figure 7), indicating that the high concentrations of L-Lys produced inhibited ASK. The relative expression of SAZ_28490 encoding ribonucleoside diphosphate reductase increased 1.54 times after ultrasonic treatment (Figure 7), which may cause increased synthesis of RDP reductase. The synthesis of RDP reductase is increased when the DNA synthesis rate in an *E. coli* cell is insufficient for the conditions of cell growth [41]. More produced ATP may be applied as a co-factor of Pls in *ε*-PL biosynthesis. Thus, the lack of ATP supply during DNA synthesis in a *S. albulus* cell may be one of the reasons for the upregulated expression of SAZ_28490. The upregulated expression of SAZ_28490 could be used as a signal to increase secondary metabolite biosynthesis at the cost of a reduction in cell growth.

The ABC transporter ATP-binding protein, encoded by SAZ_18790, affects the morphological differentiation of *Streptomyces* mycelium and the synthesis of antibiotics, and also alters the production of secondary metabolites [42]. RT-PCR further verified SAZ_18790 was upregulated by 1.47 times in the ultrasound-treated group (Figure 7). The upregulation of SAZ_18790 expression indicated that the ultrasound treatment changed the expression of membrane-transport proteins for an efflux of metabolites with the export of secondary metabolite biosynthesis. This was possibly associated with the efflux of *ε*-PL and its metabolic intermediates, causing an improvement in *ε*-PL production. Moreover, the expression of the gene SAZ_24700, which encodes serine/threonine protein kinase in the two-component system regulation (TCS) metabolic pathway [25], increased by 2.99 times (Figure 7), indicating that the gene SAZ_24700 may positively regulate the secondary metabolism of *S. albulus* [43]. 

From our experimental data, we have primarily inferred a possible increase in *ε*-PL formation following sonication. A schematic summary of our hypotheses and obtained results are displayed in Figure 8. Here, we have illustrated that the improved level of *ε*-PL was induced by the increased cell growth rate and product formation rate of *S. albulus* mycelium. In addition, the transcription level of *pls* and intracellular ATP was responsible for the enhancement of *ε*-PL biosynthesis, leading to the improvement in the content of extracellular *ε*-PL. Furthermore, the changes in mycelia viability, the activity of key enzymes, the transcription level of gene related to *ε*-PL biosynthesis and membrane-transport proteins may have contributed to the enhanced *ε*-PL yield. 

## 4. Conclusions

This study has given mechanistic insight into ultrasonic enhancement in *ε*-PL production during ultrasonic-assisted fermentation of *S. albulus*. Our results indicated that the biomass of SAR 14-116 and the concentration of *ε*-PL increased by 14.92% and 28.45% following ultrasonication (28 kHz, 0.37 W cm^−2^ and 280 W/L) relative to control. An analysis of *ε*-PL synthesis using kinetic models revealed that intense micro-convection stimulated by ultrasound/cavitation enhances the reaction velocity. In addition, ultrasound-assisted SAR14-116 fermentation may promote the synthesis of L-Lys, the precursor substance of *ε*-PL, by increasing the intracellular ATP and the metabolic intensity of glycolysis pathway, TCA cycle and DAP pathway, leading to an increase in intracellular L-Lys concentration. Furthermore, the expression level of *pls* increased after ultrasonic treatment, which directly improved the ability of SAR14-116 to synthesize *ε*-PL. These results demonstrated that ultrasound treatment is a profitable technology for *ε*-PL production, and so may be useful in metabolite fermentation from other antimicrobials.

## Figures and Tables

**Figure 1 foods-11-03525-f001:**
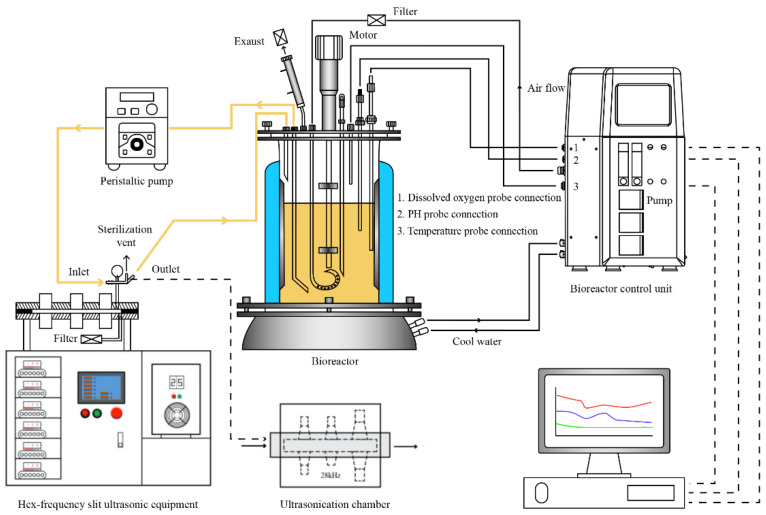
External ultrasound irradiation of broth in a recycle bioreactor.

**Figure 2 foods-11-03525-f002:**
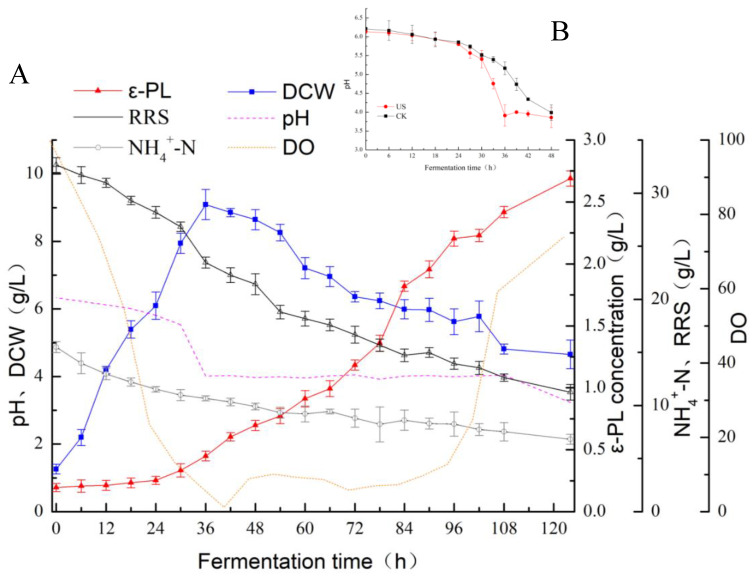
Ultrasound-assisted fermentation on SAR 14-116 production of *ε*-polylysine levels. (**A**) Time profiles of cell growth (DCW), *ε*-PL concentration (*ε*-PL), residual reduced sugar (RRS), ammonia nitrogen (NH_4_^+^-N) and dissolved oxygen (DO) under ultrasound-treated cultivation of *S. albulus* SAR 14-116; (**B**) Changes of pH between sonication treatment and non-sonication treatment.

**Figure 3 foods-11-03525-f003:**
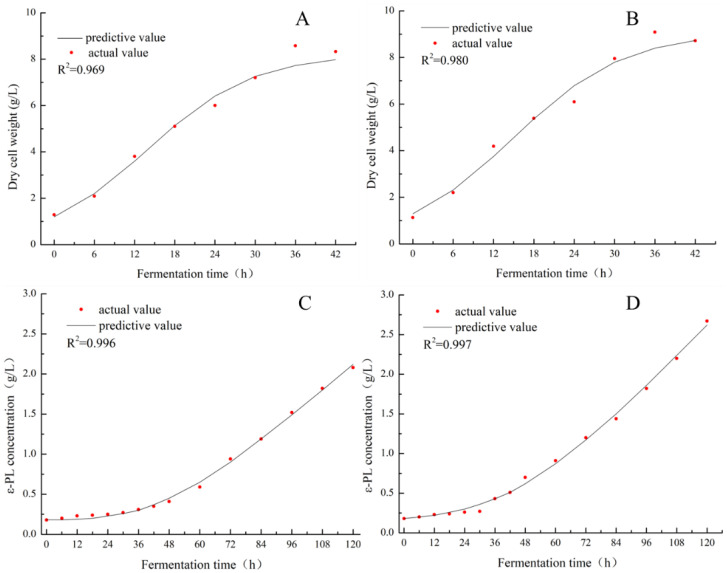
Curve fitting of kinetic model with biomass and *ε*-PL of S.albulus under ultrasound ((**B**) DCW; (**D**) *ε*-PL concentration) and without ultrasound ((**A**) DCW; (**C**) *ε*-PL concentration).

**Figure 4 foods-11-03525-f004:**
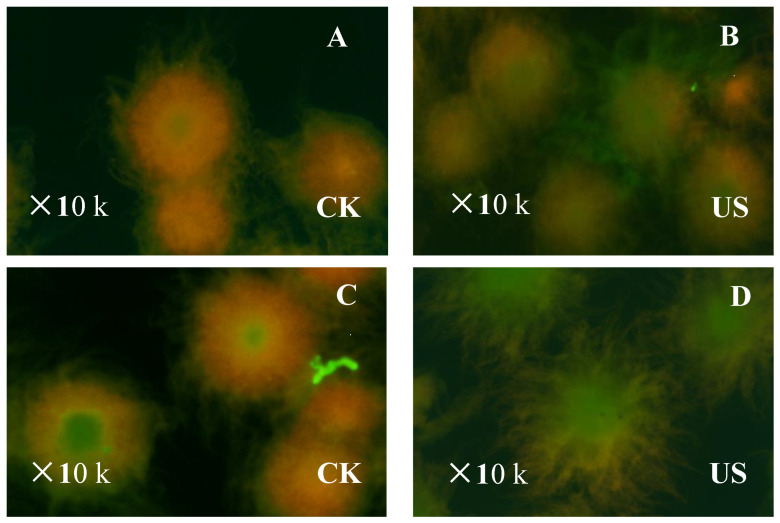
Mycelia-viability staining to observe the effect of ultrasound treatment on *S. albulus.* (Ultrasound treated for 30 min (**A**,**C**) and ultrasound treated for 60 min (**B**,**D**). Green or red means the distribution of dead and living mycelium in the cell; green means living cells with a relatively complete cell membrane, and red means dead cells with incomplete cell membrane).

**Figure 5 foods-11-03525-f005:**
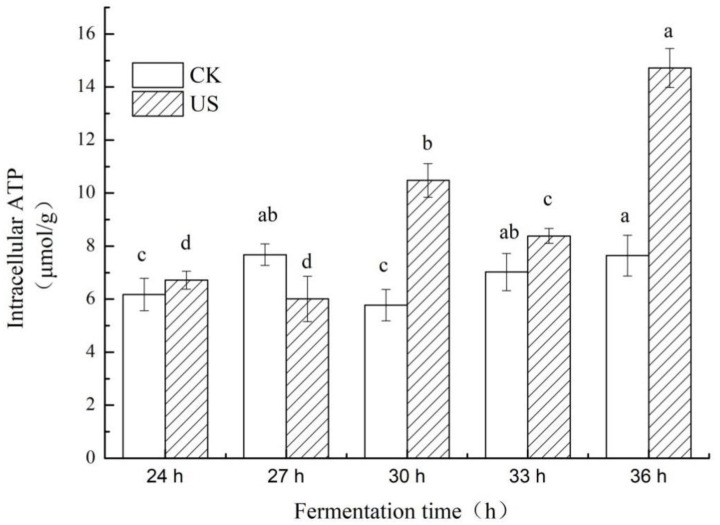
Comparison of ATP concentration during ultrasound-assisted fermentation by SAR 14-116. (Lowercase letters a, b, c and d indicate that there is a significant difference between the different culture times of fermentation (*p* < 0.05)).

**Figure 6 foods-11-03525-f006:**
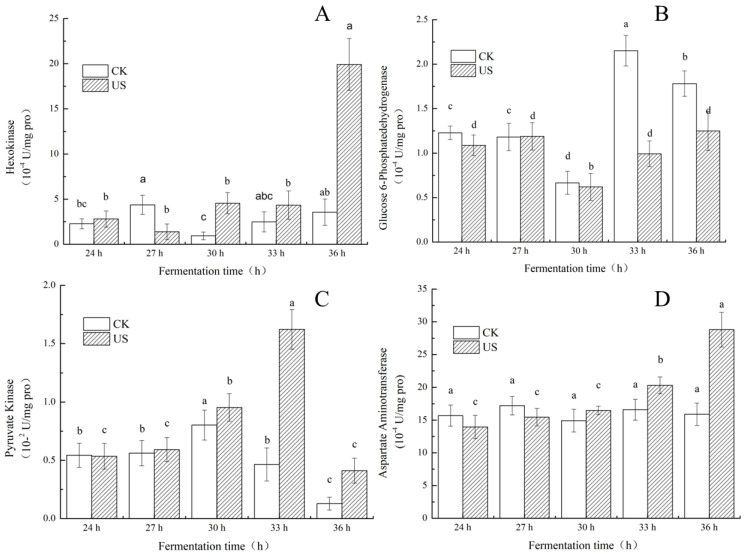
Changes of enzymatic activity of the key enzyme of ultrasound-assisted fermentation in a 3-L-jar fermenter by SAR 14-116 ((**A**) HK; (**B**) G6PDH; (**C**) PK; (**D**) AAT). (Lowercase letters a, b, c and d indicate that there is a significant difference between the different culture time of fermentation (*p* < 0.05)).

**Figure 7 foods-11-03525-f007:**
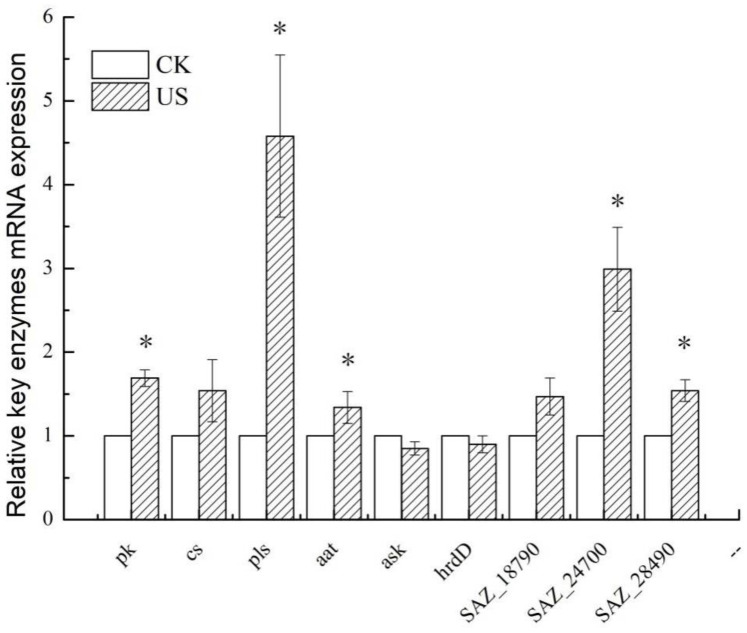
mRNA relative transcription levels of important genes for *ε*-polylysine biosynthesis (* *p* < 0.05).

**Figure 8 foods-11-03525-f008:**
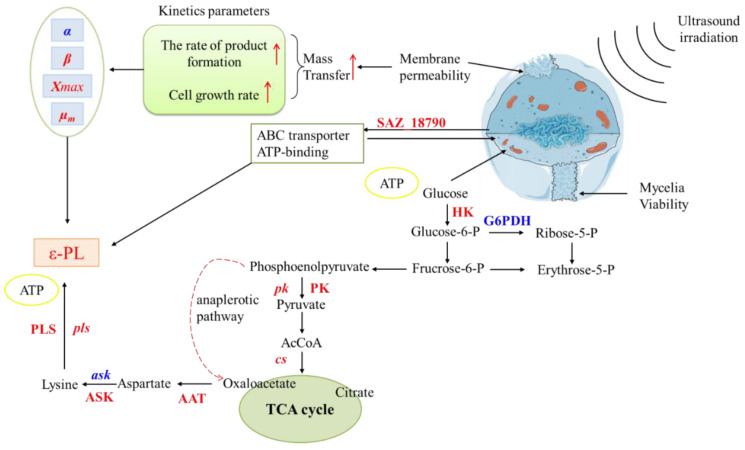
The proposed pathway of ultrasound-enhanced actinomycetes mycelia metabolism and *ε*-PL production from *S. albulus* in a 3-L-jar fermenter.

**Table 1 foods-11-03525-t001:** Respiration activity of *S. albulus* SAR 14-116 in ultrasound-assisted fermentation.

Time (min)	Ri (mmol/g·h)
0	5.38 ± 0.23 ^b^
30	6.06 ± 0.46 ^a^
60	5.41 ± 0.31 ^b^

(Lowercase letters a, b indicate that there is a significant difference between the different ultrasound-treated culture time of fermentation (*p* < 0.05)).

## Data Availability

Data are contained within the article.

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
