# Peer review of "Influence of Low-Intensity Ultrasound on ε-Polylysine Production: Intracellular ATP and Key Biosynthesis Enzymes during Streptomyces albulus Fermentation"

_foods, 2022, doi:10.3390/foods11213525_

Round 1
Reviewer 1 Report
The manuscript was well written and presented. The work is very interesting and brings insights into metabolic pathways, which are not usually seen. The discussion of the results is very good and the authors have commented on and explained the phenomenon. The manuscript has merit and needs only a few changes in the introduction and methods.
1. Introduction. Ultrasound is not new. The application of ultrasound has been done for over 30 years, so this statement is not true anymore. Please change.
2. Introduction. The introduction is contradictory. It states that ultrasound deactivates microorganisms and then later on states that can be used to increase bacteria growth. This should be better explained.
3. How do you define low-intensity ultrasound? Please give values.
4. Please define BTN and M3G in their first appearance.
5. Section 2.3 on kinetic models is badly explained. The impression is that a mathematical correlation was applied and not a phenomenological kinetic model. The model should be better explained informing how the model was constructed and the theoretical basis for it.
6. Please inform the residence time inside the ultrasonic chamber.
7. How the model parameters were obtained. Section 2.3 does not bring this information.
Reviewer 2 Report
The goal of this study was to investigate the effect of low-intensity sonication treatment on cell growth, ε-polylysine (ε-PL) yield 19 and its biological mechanism. In overall, the work was interesting and well-written. However, too old references (1984, 1994, etc.), sometimes unnecessary, were used throughout the manuscript. The references should be updated. In addition, the statistics should be checked. In the light of these, I recommend the minor revision of the manuscript.
Reviewer 3 Report
This study determines the effect of ultrasound on ε-polylysine production The introduction provides some basic information about on the possibility polylysine production by regulating fermentation processes. The objective of the study is clearly defined. The experimental apparatus is standard and is appropriate for the study. Most methods are well described and provide sufficient information to reproduce the experiments. The results are presented clearly and clearly described. The conclusions t correspond to the presented results.
minor comments.
Additional comments for Authors
In the section devoted to the method, the authors should add the intensity of the ultrasound used (in addition to the specified power density).
